# Correlation between Expression Profiles of Key Signaling Genes in Colorectal Cancer Samples from Type 2 Diabetic and Non-Diabetic Patients

**DOI:** 10.3390/life10090216

**Published:** 2020-09-22

**Authors:** Zsuzsanna Elek, Zsolt Rónai, Gergely Keszler, László Harsányi, Endre Kontsek, Zoltán Herold, Magdolna Herold, Anikó Somogyi, Zsófia Bánlaki

**Affiliations:** 1Institute of Biochemistry and Molecular Biology, Department of Molecular Biology, Semmelweis University, P.O. Box 2, H-1428 Budapest, Hungary; elek.zsuzsanna@med.semmelweis-univ.hu (Z.E.); ronai.zsolt@med.semmelweis-univ.hu (Z.R.); banlaki.zsofia@med.semmelweis-univ.hu (Z.B.); 21st Department of Surgery, Semmelweis University, P.O. Box 2, H-1428 Budapest, Hungary; harsanyi.laszlo@med.semmelweis-univ.hu; 32nd Department of Pathology, Semmelweis University, P.O. Box 2, H-1428 Budapest, Hungary; kontsek.endre@med.semmelweis-univ.hu; 4Department of Internal Medicine and Hematology, Semmelweis University, P.O. Box 2, H-1428 Budapest, Hungary; herold.zoltan@med.semmelweis-univ.hu (Z.H.); herold.magdolna@med.semmelweis-univ.hu (M.H.); somogyi.aniko@med.semmelweis-univ.hu (A.S.)

**Keywords:** colorectal carcinoma, type 2 diabetes, OpenArray, gene expression, Wnt pathway, gene interaction

## Abstract

Several lines of epidemiological and biochemical evidence support the association of type 2 diabetes mellitus (T2DM) and colorectal cancer (CRC). T2DM has been shown to impinge on the transcriptome of colon tumor cells, promoting their proliferation and invasion. In order to gain insight into diabetes-specific modulation of colon cancer signaling, we analyzed gene expression patterns of more than five hundred genes encoding signaling proteins on TaqMan OpenArray panels from colonoscopic colorectal tumor samples of type 2 diabetic and non-diabetic patients. In total, 48 transcripts were found to be differentially expressed in tumors of T2DM patients as compared to healthy colon samples. Enrichment analysis with the g:GOSt (Gene Ontology Statistics) functional profiling tool revealed that the underlying genes can be classified into five signaling pathways (in decreasing order of significance: Wnt (wingless-type)/β-catenin; Hippo; TNF (tumor necrosis factor); PI3K/Akt (phosphoinositide-3 kinase/protein kinase B), and platelet activation), implying that targeted downregulation of these signaling cascades might help combat CRC in diabetic patients. Transcript levels of some of the differentially expressed genes were also measured from surgically removed diabetic and non-diabetic CRC specimens by individual qPCR (quantitative real-time PCR) assays using the adjacent normal tissue mRNA levels as an internal control. The most significantly altered genes in diabetic tumor samples were largely different from those in non-diabetic ones, implying that T2DM profoundly alters the expression of signaling genes and presumably the biological characteristics of CRC.

## 1. Introduction

Type 2 diabetes mellitus (T2DM), a widespread metabolic disease that develops on account of relative insulin deficiency, has long been known to elevate the risk of colorectal cancer (CRC) in affected individuals [1]. Epidemiological studies conducted decades ago raised the possibility of a potential association of both diseases, and recent meta-analyses provided unequivocal statistical evidence in support of this [2,3]. Obviously, T2DM and CRC share common risk factors such as obesity, a high-fat diet, and sedentary lifestyle which have been shown to foster carcinogenesis in animal models [4]. Recent in-depth analyses led to our better understanding of molecular mechanisms underlying diabetes-related proproliferative and antiapoptotic signaling in colon epithelial cells.

T2DM is characterized by defects in pancreatic insulin secretion and impaired insulin signaling in target cells. Insulin resistance, a hallmark of T2DM, results in hyperinsulinemia eliciting elevated insulin-like growth factor 1 (IGF1) levels which promote cell growth and proliferation of mucosa cells in the large intestine [5,6]. The latter is mostly mediated by upregulation of classical mitogenic signaling pathways including MAP kinase cascades, Akt/mTOR (protein kinase B/mammalian target of rapamycin), and p21-activated protein kinase 1 (PAK1) as well as by dysregulation of principal enzymes in lipid metabolism [7,8,9]. Diabetes-specific modulation of Wnt (wingless-type)/β-catenin signaling appears to play a central role in the malignant transformation. T2DM has been shown to derepress Wnt signaling in part by downregulating the expression of Klotho, a transmembrane β-glucuronidase [10]. Following its nuclear translocation, β-catenin interacts with TCF7L2 (transcription factor 7-like 2) and transactivates several target genes including miRNA-21 [11], the long non-coding RNA KCNQ1OT1 (KCNQ1 overlapping transcript 1) [12], and Gremlin [13], factors well known to promote the proliferation, local invasion, and metastasis formation of CRC cells. Transcriptome and proteome studies revealed that the tumor-promoting potential of diabetes seems to be orchestrated by the TEAD (TEA domain family member 1) family of transcription factors and their YAP-TAZ (yes-associated protein ‒ transcriptional co-activator with PDZ-binding motif) coactivators [14].

Apart from insulin and IGF1 (insulin-like growth factor 1) signaling, there is compelling evidence for the CRC-promoting role of hyperglycemia *per se* as cancer cells require high amounts of glucose due to the Warburg effect [15]. Hyperglycemia results in the formation of advanced glycation end products (AGEs) which induce oxidative stress, inflammatory signaling, and NFκB (nuclear factor kappa B) activation [16,17]. In an attempt to uncover the factors of the pathophysiological association between T2DM and CRC, an extensive study was undertaken that shed light on hundreds of overlapped, common, and independently differentially expressed genes between diabetes and CRC. Common genes were enriched in various inflammatory pathways and in the p53 and Wnt signaling pathways [18]. Additionally, deleterious epigenetic alterations in DNA methylation patterns [19] and miRNA profiles [18] as well as decreased butyrate production by gut bacteria [20] are further factors assumed to contribute to the emergence of CRC in T2DM. Importantly, the risk of CRC in T2DM is significantly modulated by antidiabetic treatment; insulin increases, while metformin, the first-choice oral antidiabetic, reduces cancer risk by interfering with hyperactive Notch1 and mTOR signaling [20,21].

In light of the above literary data, it seems well-substantiated to assume that diabetes promotes cancer development largely by modulating intracellular signaling events. In order to gain a more global insight into the way T2DM perturbs signaling pathways in CRC, we examined the impact of T2DM on the expression of a wide array of signaling genes in CRC tissue samples. Differentially expressed genes were then ordered into functional pathways that are likely to mediate the cancer-promoting effects of diabetes in colon epithelial cells.

## 2. Results

### 2.1. Transcriptome Analysis by OpenArray

We aimed to find interactions between CRC and T2DM at the gene expression level by raising the question whether T2DM as co-morbidity can modulate CRC-specific transcriptome signatures. To this end, four sex- and age-matched patient groups from CRC patients and tumor-free individuals with or without T2DM as co-morbidity were set up. Normal (healthy control and T2DM groups) or tumor (CRC and CRC + T2DM groups) colon tissue samples were collected during colonoscopy. Total RNA was prepared and reverse transcribed and cDNA levels were quantified using a low-density OpenArray panel that simultaneously interrogated 573 transcripts involved in various signaling pathways. A heat map featuring transcript levels with unsupervised clustering showing the expression levels of each gene for each sample taken can be viewed in Appendix A. cDNA levels in the CRC, T2DM, and CRC + T2DM cohorts were normalized to those measured in the healthy control group, and possible interactions were analyzed by two-way ANOVA (analysis of variance). Table 1 presents the roster of 48 transcripts with significant statistical interaction (*p* < 0.05), that is, the expression of which was influenced both by CRC and T2DM. For 13 transcripts, the significance was even more profound (*p* < 0.01). We assume that it is these genes that mediate (and mirror) the epidemiologically confirmed association between both diseases.

Furthermore, an additional 105 transcripts were differentially expressed in tumor versus non-tumor samples (regardless of diabetic status), and 75 transcripts showed differential expression by diabetes status (irrespective of tumor status) (Appendix A). The expression of these genes was significantly altered either in diabetes or in colon cancer but was not affected by both diseases. Though these genes do not seem to contribute to and explain the role of T2DM as a risk factor of CRC, their differential expression either in CRC or T2DM might help us to better understand the pathophysiology of these diseases and find therapeutic targets.

Ordering both interacting and non-interacting genes in functional pathways provides more information on their biological relevance. We implemented the KEGG (Kyoto Encyclopedia of Genes and Genomes) database to identify relevant pathways. As can be seen in Table 2, four pathways were identified as significantly enriched by genes that showed significant correlation between CRC and T2DM status. They are (in decreasing order of significance) the Wnt (wingless-type), Hippo, TNF (tumor necrosis factor), and PI3K/Akt (phosphoinositide-3 kinase/protein kinase B) signaling cascades. In other words, T2DM status apparently interferes with the mechanism of how these four pathways are altered in CRC. Thus, these pathways could be of fundamental importance to resolve the potential modulatory role of T2DM in CRC pathogenesis. Unexpectedly, a collagen and thrombin-induced platelet activation pathway was also unveiled with marginal significance, which might shed light on a differential regulatory role of normal and diabetic platelets in the vascular dissemination of colon cancer cells. We also performed KEGG pathway analysis for signaling genes with tumor-specific (panel A) and diabetes-specific (panel B) expression alterations (Appendix A).

### 2.2. Validation of Expression Data by qPCR

In order to reaffirm the pathophysiological relevance of genes shown in Table 1 in mediating the tumor-promoting effect of diabetes, transcript levels were recapitulated on independent CRC samples. Six non-diabetic and six sex- and age-matched type 2 diabetic patients having undergone abdominal CRC surgery were involved in this study. Tissue samples were taken both from the territory of the tumor as well as from the surrounding cancer-free area of each operative specimen. cDNA levels of 23 genes with statistically significant interaction were tested by individual qPCR assays, and intra-tumor expression levels were normalized to those measured in the surrounding area. Genes with the most significant and/or most profound OpenArray expression differences were preferably chosen. This approach has the advantage of precluding individual differences in gene expression by using the tumor-free tissue from the same person, allowing us to focus on gene expression changes attributed solely to the presence of T2DM. A heat map of the 23 genes assayed is shown in Appendix A. Gene expression data were analyzed by a paired t-test. Four genes were found with significantly different expression levels in the context of tumor vs. surrounding, ostensibly normal tissue in diabetic and non-diabetic cohorts, respectively (Table 3, highlighted in bold). It is noteworthy that these 4-strong batteries comprised different genes with the exception of CSF1R which was present in both but with markedly different significance (0.0016 vs. 0.0478). These results thus also argue that type 2 diabetes correlates with gene expression profiles and thereby tumor signaling pathways in colorectal adenocarcinoma.

## 3. Discussion

A large body of epidemiological and experimental evidence argues that type 2 diabetes is undoubtedly a risk factor in colorectal carcinogenesis, though exact molecular mechanisms remain largely elusive. As cancer-associated signaling pathways are indispensable to drive proliferation and migration of tumor cells, it seemed reasonable to perform a comprehensive expression analysis of signaling genes to reveal diabetes-specific transcriptomic signatures in CRC. Pathway analysis of data presented in Table 1 revealed that the Wnt signaling cascade was most significantly affected as eight genes of this pathway were differentially expressed in tumors of diabetic individuals compared to those in non-diabetic patients (Table 2). Of them, five genes (AXIN1, CTBP2 (C-terminal-binding protein 2), LRP5 (low-density lipoprotein receptor-related protein 5), MMP7 (matrix metalloproteinase 7), MAPK9 (mitogen-activated protein kinase 9) were downregulated while the expression of three genes (F2D7 (Frizzled 7), WNT7A, and 16) was enhanced by diabetes. Nevertheless, the pattern of the interaction was quite unusual in case of the F2D7 gene as its expression was diminished both by T2DM and CRC alone, while a twofold elevation in mRNA levels was found in the tumors of diabetic subjects (Table 1). Another peculiar interaction pattern was detected with MMP7 as its expression was reduced by T2DM in the tumor, while diabetes alone slightly upregulated that gene in normal gut epithelial cells. These untypical correlations imply that diabetes differentially modulates signaling pathways in the tumor and in healthy mucosa cells of the large intestine.

The lion’s share of gene expression modulation we detected in Wnt signaling in CRC cells bears pathophysiological relevance in light of the established role of diabetes as a tumor-promoting condition [1,2]. The Wnt morphogenic pathway is essential in cell fate specification and body axis patterning during embryogenesis, but it also supports proliferation and migration of CRC cells. In fact, Wnt signaling is often aberrantly activated in CRC due to loss-of-function mutations in the gene encoding APC (adenomatous polyposis coli), a key component of the destruction complex that normally degrades β-catenin [22,23]. In the presence of T2DM, however, upregulation of this pathway seems to be more pronounced as expression of the Frizzled 7 membrane receptor (F2D7) and its ligands WNT7A and WNT16 is increased, while that of negative regulators such as AXIN1, a proapoptotic scaffold protein of the β-catenin destruction complex and CTBP2, a corepressor preventing β-catenin from activating its target genes, is suppressed. Though the overexpression of Wnt ligands and receptors does not seem to confer further growth advantage with respect to APC-negative tumors, downregulation of CTBP2 that acts downstream of APC and activation of non-canonical Wnt signaling pathways might confer biological relevance in our findings.

These data confirm recent findings of Ivonne Wence-Chavez et al. [24], who performed a global transcriptome analysis in a setting similar to ours and found that transcript levels of WNT3A, LRP6, TCF7L2, and FRA-1 (Fos-related antigen 1) were significantly higher in CRC cells of diabetic individuals compared to non-diabetic controls. Of these genes, WNT3A and LRP6 were also included on our OpenArray panel. Although we did not find these genes differentially expressed in our samples, the modulation of two alternative Wnt ligand isoforms (WNT7A and 16) and the LRP6 isoform LRP5 as well as a number of auxiliary members of the canonical Wnt pathway provides convincing evidence that the CRC-promoting effect of insulin resistance might at least partly be mediated by heightened Wnt signaling. The reason why we detected other genes of the Wnt cascade might be due to methodological (microarray vs. OpenArray) and genetic differences (Mexican vs. Caucasian).

Wnt activation underlying T2DM-related CRC was hypothesized by Jin already in 2008 [25]. Two years later, Sun et al. demonstrated that insulin induces some members of the Wnt signalling pathway in a rat intestinal cell line [26]. Li et al. showed that non-transformed colorectal epithelial cells accumulate more β-catenin and undergo faster proliferation in T2DM patients, further arguing for the critical role of the Wnt pathway in CRC initiation [27]. As Wnt inhibition is a burgeoning field in targeted chemotherapy research [28], our results might be of relevance for the pharmacological therapy of CRC patients with type 2 diabetes.

Though we identified four other relevant pathways as well, the statistical power of significance of these associations was at least an order of magnitude poorer than that of the Wnt signal transduction. Of them, the Hippo tumor suppressor pathway is known to control organ size by subtly integrating proliferative and apoptotic signals. Prevention of tissue overgrowth is achieved by Hpo (hippo) - and Wts-mediated inhibition of Yorkie, a transcriptional coactivator proto-oncogene [29]. Apart from promoting tumor growth via upregulation of a gene battery largely overlapping with that of Wnt signaling (Table 2), our data suggest that coordinated activation of the Hippo and PI3k/Akt pathways in T2DM might lead to exuberant extracellular matrix production and fibrosis in colorectal malignancies. This can be assumed from the simultaneously increased expression of two collagen genes (COL1A1 (alpha-1 type I collagen) and COL1A2), the SERPINE1 gene encoding plasminogen activator inhibitor-1 (a serine protease inhibitor) and CTGF (connective tissue growth factor), and downregulation of matrix metalloproteinase 7 (MMP7) in diabetes (Table 1). In fact, overexpression of SERPINE1 and CTGF has been reported in many cancer types including CRC. Perhaps it might be of considerable interest to see whether a histopathological study could confirm this assumption.

AREG, coding for amphiregulin, a member of the epidermal growth factor family, is reportedly overexpressed in colon cancer [30] and activates both MAP kinase and PI3K/Akt canonical signaling in tumor cells. Remarkably, it is part of the Hippo and PI3K signaling network and exhibits a more than fourfold overexpression in diabetes and CRC.

Last but not least, activation of the TNF signaling pathway—among others—is marked by enhanced expression of genes representing leukocyte migration and recruitment. The product of the CCL2 (chemokine (C – C motif) ligand 2) gene is monocyte chemoattractant protein 1 (MCP1), while SELE (selectin E) and ICAM1 (intercellular adhesion molecule 1) give rise to cell adhesion molecules serving the extravasation of proinflammatory mono- and granulocytes. As inflammation is one of the enabling characteristics of malignant tumors [31], these gene expression aberrations appear to accelerate the acquisition of the malignant phenotype of CRC.

The second part of the study was carried out in order to gain further evidence in support of the hypothesis that gene expression signatures in CRC are greatly affected by T2DM. To preclude the potentially perplexing impact of individual genetic variations, gene expression patterns of 23 selected genes were compared in surgically removed and histologically verified colorectal tumors, using the surrounding, apparently normal, tissue as a control. Data summarized in Table 3 unambiguously show that genes with the most significantly altered expression in diabetes-free tumors (CDK2 (cycline dependent kinase 2), CDC25A (cell division cycle 25 homolog A), DUSP14 (dual specificity phosphatase 14), and CSF1R (colony stimulating factor 1 receptor) were largely different from those in diabetes-associated CRC (CSF1R, PCK2 (phosphoenolpyruvate carboxykinase 2), BMP5 (bone morphogenic protein 5), and SMAD3 (SMAD family member 3). CSF1R was present in both gene sets but with very different significance.

Upregulation of CDK2, the CDC25A phosphatase and DUSP14 in diabetes-free tumors is consistent with the tumor phenotype as these genes are reportedly overexpressed to stimulate cell proliferation and invasion in CRC [32,33,34]. Suppression of CSF1R expression both in diabetes-free and especially in diabetes tumors seems to be counter-productive for the malignancy via decreasing the influx of immunosuppressive tumor-associated macrophages (TAM) which in turn compromise antitumor activities of CD8+ cytotoxic lymphocytes [35]. It is also hard to reconcile downregulated PCK2 and SMAD3 transcript levels in tumors from diabetic patients with more vigorous tumor propagation seen in T2DM as PCK2 and SMAD3 have been shown to foster tumor growth and dissemination [36,37]. On the other hand, reduced mRNA levels of the BMP5 tumor suppressor gene appear to support the notion that T2DM fosters CRC progression [38].

From a technical point of view, our qPCR-based validation of OpenArray data was firm and sound as the tendencies of significant gene expression alterations in individual qPCR assays (Table 3) run on surgical specimens were the same as those measured by OpenArray on colonoscopic samples. Nevertheless, the apparently anomalous expression levels of some genes imply that a systems biology approach, i.e., global survey of multiple genes and subsequent pathway analysis, is a more reliable tool in answering biological questions than studying expression levels of single genes. Small sample size was another limitation of the study, which undoubtedly reduced the statistical power of the results and also limited the extent of analyses that could be conducted. The present sample size did not allow adjustment for clinical parameters potentially affecting gene expression levels, such as tumor staging or location, or assessment of a possible relationship between diabetes status and cancer severity. Importantly, variability in other parameters that can exert influence on gene expression levels most probably also reduced the statistical power and hindered the detection of smaller effects between groups. Although subjects involved in this study were carefully selected for homogeneity in co-morbidities and medication, random variance in other factors, most importantly cellular composition of sample tissues and consensus molecular subtypes, might also have affected the results. It is also crucial to point out that protein expression patterns do not necessarily correspond to mRNA levels, and subcellular localization as well as enzyme activity may also vary, influencing protein functions. These limitations underline the necessity of further studies performed on larger and well-characterized sample sets, investigating protein levels as well as function.

## 4. Materials and Methods

### 4.1. Colonoscopy Samples

The first part of the study was based on a set of patients undergoing colonoscopy due to gastrointestinal symptoms at the Department of Internal Medicine and Hematology, Semmelweis University, Budapest, Hungary. The diagnosis of colorectal cancer (CRC) was set up by histopathological examination of excised tissue samples. Only verified tumor tissue samples were obtained from CRC patients and processed for gene expression analysis. The diagnosis of type 2 diabetes (T2DM) among these patients had been set up according to current international recommendations [39] and they were receiving proper oral antidiabetic medication (metformin) to ensure HbA1C levels below 8% at the time of tissue sampling. Subjects with autoimmune, metabolic (other than diabetes), infectious or inflammatory bowel diseases as well as those receiving immunosuppressive drugs were excluded from this study. Subjects meeting inclusion criteria were divided into four sex- and age-matched cohorts based on their colorectal carcinoma and type 2 diabetes status. The healthy (non-tumor, non-diabetic) control group contained two men and four women (mean age ± SD: 68 ± 6.5 years), the tumor-free but diabetic group included three men and three women (mean age ± SD: 63 ± 4.5 years), the non-diabetic group with CRC comprised three men and two women (mean age ± SD: 63 ± 7.4 years), while the CRC + T2DM group consisted of two men and three women (mean age ± SD: 65 ± 4.9 years). Both the diabetic and the non-diabetic CRC groups comprised one subject with T2 and four subjects with T3 tumor size according to the TNM staging system. Two subjects had N0 and three subjects had N1 lymph node status in the non-diabetic CRC group while three subjects had N0 and two subjects had N1 stage in the diabetic CRC group. With regard to metastasis status, four subjects had stage M0 and one subject had M1 stage disease in both CRC groups. In the non-diabetic CRC group, one person had right-sided and four patients had left-sided tumors. In the diabetic CRC group, three subjects had right-sided and two subjects had left-sided tumors. Tissue samples collected during colonoscopy were preserved in RNAlater Stabilization Solution (ThermoFisher Scientific, Waltham, MA, USA) and stored at −20 °C until RNA isolation.

### 4.2. Surgically Removed Samples

The second part of the study was based on an independent cohort of CRC patients undergoing surgical resection at the 1st Department of Surgery, Semmelweis University, Budapest, Hungary. Intraoperative samples were collected both from the tumor and from the adjacent normal tissue in all patients. Of them, a total of six non-diabetic and six T2DM patients were included in this study. Both groups comprised three men and three women. Mean age ± SD was 69 ± 11.5 years and 70 ± 10.4 years in the non-diabetic and the T2DM group, respectively. Only patients with primary colorectal adenocarcinoma were included. Exclusion criteria included autoimmune and metabolic diseases, immunosuppressive therapy, and neoadjuvant radio- and chemotherapy. In the non-diabetic group, two subjects had T2, three subjects had T3 and only one subject had T4 tumor size under the TNM staging system. Four subjects had N0 and two subjects had N1 lymph node status, and five subjects had M0, and a single subject had M1 metastasis status. In the diabetic group, five subjects had T3 and one subject had T4 tumor size. Two patients had N0 and four participants had N1 lymph node status, while all six subjects had M0 metastasis status. In the non-diabetic group, three subjects had right-sided and three subjects had left-sided tumors. In the diabetic group, two people were diagnosed with right-sided and four subjects with left-sided tumors. Tissue samples were removed at the 2nd Department of Pathology, Semmelweis University, Budapest, Hungary. They were preserved in RNAlater Stabilization Solution and stored at −20 °C until further use.

### 4.3. RNA Extraction and cDNA Synthesis

Total RNA was isolated using the PureLink^®^ RNA Mini Kit (ThermoFisher Scientific, Waltham, MA, USA). mRNA was reverse transcribed into cDNA with the SuperScript VILO cDNA Synthesis Kit (all reagents were obtained from ThermoFisher Scientific (Waltham, MA, USA)).

### 4.4. OpenArray Analysis of Colonoscopic Tissue Samples

Equivalent amounts of two cDNA samples were pooled in each cohort so that three pools for each of the four colonoscopy groups were created for gene expression measurement (in the case of the tumor + non-diabetes and the tumor + diabetes groups which involved only 5-5 subjects, two pools in duplicate and one individual sample were measured). Gene expression was quantified by real-time PCR using the TaqMan™ OpenArray™ Human Signal Transduction Panel for QuantStudio™ 12K Flex (ThermoFisher Scientific, Waltham, MA, USA). Reactions were performed in triplicate. Expression levels were normalized to the geometrical mean of the 24 internal reference genes included in the panel.

### 4.5. Quantitative Real-Time PCR Assays on Intra-Operative Samples

Individual, predesigned TaqMan gene expression assays (ThermoFisher Scientific, Waltham, MA, USA) were performed on intraoperative samples to recapitulate and validate the significantly different expression of 23 out of 48 genes found in the colonoscopy samples. The final gene set for individual testing was carefully set up using the following inclusion criteria: (1) key genes belonging to separate signaling cascades and (2) genes with the clearest inter-group expression pattern differences were selected. Assays were run in triplicate on a QuantStudio™ 12K Flex system. Here, gene expression levels were normalized to the geometrical mean mRNA levels of the CASC3 (assay: Hs00201226_m1) and POLR2A (assay: Hs00172187_m1) internal control genes.

### 4.6. Statistical and Computer Analyses

Statistical analyses were carried out using Statistica software version 13.4.0.14 (TIBCO Software Inc, Palo Alto, CA, USA, http://tibco.com). Two-way ANOVA was used for assessing the possible combined and/or separate effect of the presence of CRC and/or T2DM on gene expression in the pooled colonoscopy samples. Comparison between normalized mRNA levels in the intraoperative tumor and surrounding tissues was performed by a two-tailed paired t-test. The significance level was set at alpha = 0.05.

Optimal reference genes to quantify gene expression levels by individual TaqMan assays were determined by qBase+ software version3.2 (Biogazelle, Zwijnaarde, Belgium) (https://www.qbaseplus.com/) [40], based on results obtained with the 24 internal reference assays included on the OpenArray^®^ plate.

Heat maps were generated with the Heatmapper web server developed by the Wishart Research Group at the University of Alberta, Canada (http://www.heatmapper.ca/) [41]. Dendrograms were created with the Euclidean distance measurement method applying average linkage clustering.

Enrichment analysis for gene sets showing significant inter-group differences was performed using the g:GOSt functional profiling tool of g:Profiler online software (version e100_eg47_p14_7733820; developed and maintained at the University of Tartu, Estonia, at the Institute of Computer Science, Bioinformatics, Algorithmics and Data Mining Group BIIT) (http://biit.cs.ut.ee/gprofiler/) [42], based on terms defined by the Kyoto Encyclopedia of Genes and Genomes (KEGG) database (release 95.0, developed by Kanehisa Laboratories, Kyoto, Japan and Tokio, Japan) (https://www.genome.jp/kegg/) [43,44] as a reference.

### 4.7. Ethics Approval

Participation in this study was on a voluntary basis. All participants gave written informed consent. The study was approved by the Regional and Institutional Committee of Science and Research Ethics of Semmelweis University (permission No.: 39/2014) and by the ETT-TUKEB ethics board (permission No.: 8573-9/2017/EÜIG). The study was conducted in concordance with the WMA Declaration of Helsinki; handling of patient data was in accordance with the General Data Protection Regulation issued by the European Union.

## Figures and Tables

**Table 1 life-10-00216-t001:** Transcripts of significant interaction between tumor and diabetes status.

Gene	Fold Change Diabetes	Fold Change Tumor	Fold Change Diabetes + Tumor	*p* Value Interaction	*p* Value Tumor	*p* Value Diabetes
MMP7	1.44	194.67	34.65	0.0013	0.0001	0.0014
COL1A1	0.76	1.73	7.88	0.0018	0.0005	0.0029
*CYP19A1* ^1^	0.69	0.98	2.76	0.0019	0.0021	0.0129
DUSP14	0.85	1.35	4.96	0.0021	0.0007	0.0034
COL1A2	0.56	1.03	4.22	0.0023	0.0021	0.0105
CD86	0.60	0.55	0.98	0.0031	0.7203	0.8434
SLC44A2	0.94	0.96	0.36	0.0034	0.0015	0.0011
PCK2	1.09	0.87	0.34	0.0039	0.0005	0.0209
CTGF	0.82	0.78	2.48	0.0054	0.02	0.0161
*CYP19A1* ^2^	0.50	1.22	2.36	0.0078	0.0021	0.2031
FZD7	0.82	0.44	1.99	0.0082	0.2527	0.025
CCL2	0.99	1.35	4.23	0.0091	0.0028	0.0093
S100A12	0.23	3.78	71.99	0.0093	0.0062	0.0104
CDC25A	0.85	2.15	1.19	0.0101	0.0003	0.0017
CSF1R	0.66	0.33	0.54	0.0106	0.0014	0.4286
AXIN1	0.91	1.36	0.64	0.0154	0.7045	0.0046
RGS2	0.80	1.01	2.36	0.0156	0.0144	0.0537
WNT7A	1.37	1.98	9.24	0.0175	0.005	0.0107
STAT6	1.06	0.96	0.36	0.0179	0.0097	0.0433
TLR2	0.53	0.66	2.12	0.0191	0.0951	0.1729
LPAR2	1.21	1.25	0.54	0.0206	0.2194	0.1638
PTPRC	0.62	0.69	1.05	0.0212	0.6517	0.9565
WNT16	1.21	1.53	5.82	0.0221	0.0084	0.015
TLR6	0.69	0.59	0.96	0.0222	0.5499	0.8096
PLAU	0.82	1.08	6.21	0.0237	0.021	0.0317
COL3A1	0.46	0.80	1.85	0.0251	0.073	0.4087
TP53I3	1.13	0.89	0.51	0.0264	0.0049	0.2134
TNFAIP3	0.75	0.81	1.80	0.0282	0.1021	0.1529
ELK4	0.95	1.44	0.81	0.0287	0.2073	0.0148
CDK2	0.94	2.46	1.39	0.029	0.001	0.0183
DRD2	1.37	1.09	0.50	0.0306	0.0651	0.5616
CTBP2	0.81	1.47	0.55	0.031	0.46	0.004
ICAM1	0.74	1.05	1.93	0.032	0.0226	0.1895
*SELE* ^3^	0.27	5.19	19.93	0.035	0.0045	0.0507
CDKN1C	0.80	0.48	2.04	0.0351	0.3279	0.086
AREG	1.29	1.59	4.30	0.0355	0.0056	0.0142
SH2B1	1.02	0.97	0.40	0.0381	0.0257	0.0528
F2R	0.89	0.99	2.36	0.0385	0.0404	0.0698
*SELE* ^4^	0.69	2.26	8.03	0.0399	0.0085	0.0591
CEBPB	1.17	2.08	3.87	0.0414	0.0005	0.019
PYGO1	0.59	0.69	1.00	0.0415	0.7447	0.7434
MAPK9	0.95	1.28	0.82	0.0423	0.4091	0.0169
MMP10	0.21	4.91	18.28	0.0429	0.0058	0.0651
GNAS	0.89	1.53	0.78	0.0449	0.1465	0.0118
LRP5	0.79	1.34	0.50	0.0472	0.8559	0.0044
PPARG	0.89	1.10	0.47	0.0489	0.1954	0.0111
SERPINE1	0.91	0.81	15.57	0.0494	0.054	0.0516
PMAIP1	0.92	2.07	3.35	0.0499	0.0004	0.0762

Results from the colonoscopy samples interrogated by the OpenArray™ Human Signal Transduction Panel are shown. Only transcripts with significant levels of interaction between tumor and diabetes status (*p* ≤ 0.05 based on 2-way ANOVA (analysis of variance)) are shown. Relative expression levels are compared to control (tumor-free, non-diabetic) samples and normalized for 24 housekeeping assays included in the panel. Genes whose expression was quantified with more than one TaqMan^®^ assay (with different primers and probes) in the panel are shown in italics. In these cases, precise assay identifiers are referred to with upper index figures as follows: upper indexes 1 (^1^) and 2 (^2^) refer to *CYP19A1* assays Hs00903413 and Hs00903411, respectively, while upper indexes 3 (^3^) and 4 (^4^) refer to *SELE* assays Hs00174057_m1 and Hs00950401_m1, respectively.

**Table 2 life-10-00216-t002:** KEGG database terms from the enrichment analysis of genes whose expression showed a significant correlation between tumor and diabetes status.

KEGG Database Main Category	KEGG Database Subcategory	KEGG Term	Adjusted *p* Value *	Query Genes Matching KEGG Terms
3. Environmental Information Processing	3.2 Signal transduction	Wnt signaling pathway	0.000172	AXIN1, CTBP2, FZD7, LRP5, MAPK9, MMP7, WNT16, WNT7A
3. Environmental Information Processing	3.2 Signal transduction	Hippo signaling pathway	0.001594	AREG, AXIN1, CTGF, FZD7, SERPINE1, WNT16, WNT7A
3. Environmental Information Processing	3.2 Signal transduction	TNF signaling pathway	0.002343	CCL2, CEBPB, ICAM1, MAPK9, SELE, TNFAIP3
3. Environmental Information Processing	3.2 Signal transduction	PI3K-Akt signaling pathway	0.009072	AREG, CDK2, COL1A1, COL1A2, CSF1R, F2R, LPAR2, PCK2, TLR2
5. Organismal Systems	5.1 Immune system	Platelet activation	0.04599	COL1A1, COL1A2, COL3A1, F2R, GNAS

KEGG (Kyoto Encyclopedia of Genes and Genomes) database categories and terms were determined by the g:Profiler tool. Only terms with presumed biological relevance, that is, matching KEGG categories 3 (Environmental Information Processing), 4 (Cellular Processes) or 5 (Organismal Systems) are indicated. * Adjustment for multiple testing was performed with the g:SCS algorithm of g:Profiler, suitable for the analysis of hierarchically related terms.

**Table 3 life-10-00216-t003:** Gene expression differences between the tumor and its surrounding tissues from diabetic and non-diabetic patients.

Panel A (Diabetic Group)			Panel B (Non-Diabetic Group)		
Gene	*p* Value (Tumor Tissue vs. Surrounding Tissue)	Fold Change * (Tumor Tissue vs. Surrounding Tissue)	Gene	*p* Value (Tumor Tissue vs. Surrounding Tissue)	Fold Change * (Tumor Tissue vs. Surrounding Tissue)
**CSF1R**	0.0016	0.41	**CDK2**	0.0025	1.69
**PCK2**	0.0091	0.44	**CDC25A**	0.0454	1.71
**BMP5**	0.0127	0.16	**DUSP14**	0.0464	3.03
**SMAD3**	0.0320	0.48	**CSF1R**	0.0478	0.52
SLC44A2	0.1298	(-)	SMAD3	0.1151	(-)
COL1A1	0.1372	(-)	BMP5	0.1354	(-)
PTPRC	0.1434	(-)	MMP7	0.1407	(-)
DUSP14	0.2114	(-)	AXIN1	0.1998	(-)
MMP7	0.2196	(-)	COL1A1	0.2182	(-)
STAT6	0.2412	(-)	FZD7	0.2393	(-)
TGFB3	0.2640	(-)	CYP19A1	0.2662	(-)
S100A12	0.3444	(-)	ELK4	0.2985	(-)
ELK4	0.3513	(-)	S100A12	0.3458	(-)
CYP19A1	0.3795	(-)	SLC44A2	0.3464	(-)
CDK2	0.4073	(-)	CTGF	0.3799	(-)
AREG	0.4897	(-)	RGS2	0.3861	(-)
AXIN1	0.5205	(-)	AREG	0.4137	(-)
FZD7	0.5296	(-)	CCL2	0.4637	(-)
CTGF	0.5998	(-)	TGFB3	0.5355	(-)
RGS2	0.6201	(-)	CD86	0.6652	(-)
CCL2	0.6588	(-)	PCK2	0.6942	(-)
CDC25A	0.7379	(-)	STAT6	0.7104	(-)
CD86	0.9452	(-)	PTPRC	0.7532	(-)

Dependent *t*-test scores were calculated from expression levels of surgically removed tissue samples. Relative expression levels were generated by dividing tumor expression levels with those of the surrounding tissue and normalized to the geometrical mean of *CASC3* (CASC3 exon junction complex subunit) and *POLR2A* (RNA polymerase II subunit A) internal control transcript levels as recommended by the qBase+ software version3.2 (Biogazelle, Zwijnaarde, Belgium). Fold changes are indicated only where significant (*p* ≤ 0.05) and are highlighted in bold. Panel A: gene expression differences between tissues in the diabetic group; panel B: gene expression differences between tissues in the non-diabetic group.

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
