# Peer review of "Correlation between Expression Profiles of Key Signaling Genes in Colorectal Cancer Samples from Type 2 Diabetic and Non-Diabetic Patients"

_life, 2020, doi:10.3390/life10090216_

Round 1
Reviewer 1 Report
To gain an insight into the association between type 2 diabetes mellitus (T2DM) and increased risk of colorectal cancer (CRC), the authors looked at the expression of 573 transcripts in colonoscopic colorectal tissues of CRC, T2DM, CRC+T2DM, and healthy cohorts, and found Wnt, Hippo, TNF, and PI3K/Akt signaling are likely to mediate the CRC-promoting effect in T2DM patients. These differentially expressed genes were further analyzed in tumors surgically removed from CRC patients with or without diabetes. The results indicate T2DM dramatically altered gene expression profiles that are likely responsible for promoting CRC. The paper is potentially interesting, highlighting possible therapy or preventive medicine targeting these pathways for CRC patients with T2DM. The issues described below need to be clarified before the paper is suitable for publication are detailed below.
The expression of the transcripts doesn’t necessarily reflect the expression of their corresponding proteins. The authors, therefore, need to confirm the protein expression in tumor tissues. I would recommend the authors to perform immunohistochemistry for proteins (the transcripts with p < 0.05) listed in Table 3. This will make the authors’ statement more convincing.
Reviewer 2 Report
Clearly written manuscript examining the interplay/influence of type 2 diabetes mellitus (T2DM) and colorectal cancer (CRC).
The study replies on the use of TaqMan® Gene Expression Assay "OpenArray" to examine the expression of 573 transcripts within 4 groups of samples, n=5 or 6 per group. The expression of a subset of 23 genes was further validated in independent samples, n=6 per group, using individual TaqMan qPCR assays.
Points to address:
It would be good for the authors to acknowledge the small sample sizes and the limitations this has on the study.
In lines 81/82 of the Manuscript "Tissue samples were collected” and in section 4.1 Colonoscopy samples - it would be beneficial to clarify / categorically state that the CRC group, and the CRC+T2DM group include only cancerous tissues. It isn't clear, as written, whether the colonoscopy procedure collected cancerous and/or normal tissues from these patient groups.
The formatting in lines 104, 105 +106 needs to be revised, such that the probe names read as a sentence rather than a disjointed list as currently presented.
Clarity for the rational for the selection of the 23 genes in the second section of the study would help. These seem to bear no direct correlation with the 48 transcripts identified by the OpenArray as showing significant statistical interactions.
The clinical staging/grade and anatomical location of the tumours analysed should be described if possible. It is known that gene expression levels and patient outcome are influenced by these two important factors, and without these, it is not possible to exclude other factors giving rise to the findings that are independent of T2DM status.
Reviewer 3 Report
Type 2 diabetes mellitus (T2DM) and colorectal cancer (CRC) are effecting many people worldwide causing high morbidity and mortality. Based on some evidence showing an association between occurrence of T2DM and CRC, Elek et al. set out to get insights into possible T2DM-specific transcriptional modulation of CRC cells. The authors perform a TaqMan OpenArray on colon and CRC tissue biopsied from different cohorts of people: 6 healthy individuals, 6 T2DM patients (without CRC), 5 T2DM patients with CRC and 5 CRC patients (without T2DM). 48 transcripts were shown to be significantly expressed in CRC and T2DM tissue samples in comparison to healthy colon tissue samples. Next, the authors performed gene set enrichment analysis to identify signalling pathway altered in CRC tissue of patients also diagnosed with T2DM in comparison to the CRC tissue of T2DM-free patients. Lastly, the authors obtained samples from both CRC tissue and surrounding tissue from each 6 T2DM and non-T2DM patients. Gene expression analysis revealed limited genes, which were differentially expressed between CRC tissue and surrounding tissue in the two cohorts. Only one gene (CSF1R) was found to be differentially expressed in both cohorts.
Overall, the manuscript is well written. However, the study lacks originality, significance, quality of presentation and therefore will be of limited interest to the readers. My major concerns are outline below.
1) Headline: The headline is misleading, as the study only shows a correlation between gene expression profiles of CRC with and without T2DM co-diagnosis. No functional experiments are presented to demonstrate a causation between T2DM and CRC, therefore the use of the term "modulation" is not justified. Furthermore, the authors used CRC biopsies, which do not only contain epithelial cancer cells. Please choose a title that is in line with the data presented.
2) Methodology
(a) The samples biopsied contain not only epithelial (cancer) cells, but also other cells of the tumour microenvironment. This is evident in the gene list presented in table 1: PTPRC encodes CD45, which is a prominent marker of haematopoietic cells, TLR2 and TLR6 are also robustly expressed by haematopoietic cell subpopulations, PPARG is broadly expressed by endothelial cells and ICAM1 is a markers of both endothelial and haematopoietic cell lineages. Many of these genes are also listed in Table 3. This is a rather important issue, as this may account for large heterogeneity in cell composition between patients causing strong variation in gene expression patterns. Please take this limitation into account when analysing and discussing the data.
(b) CRC can be subdivided into four consensus molecular subtypes (CMS), which show large differences in their mutational status, gene expression pattern and diagnosis/prognosis (see Guinney et al., 2015, Nature Medicine). However, the authors used a very low number of patients 10 in total with each 5 CRC patients with and without T2DM co-diagnosis. Therefore, a lot of the heterogeneity in the transcriptional profiles may be explained by the CMS of the individual CRC samples. Please provide a proper pathological and genetic diagnosis for each of the CRC patients to account for this matter. Please make sure that the samples are not only matched to the sex and age of the CRC patients, but also their CMS status to avoid ambiguity.
3) Data analysis, presentation and interpretation
(a) Overall, data presentation must be significantly improved. It is not valid to present the data in Table 1 as fold changes over the control samples, as these controls were not obtained from the same CRC and/or T2DM patients. Please generate heat maps incorporating the gene expression profiling with unsupervised clustering showing the expression levels of each gene for each sample taken, as it is standard in the field for presenting gene-expression profiles. Another option would be to show the expression levels for each gene per sample cohort in column charts displaying standard deviations or standard error of the mean as well as the individual data points (as dots). Please also amend Table 3 accordingly.
(b) Table 2 shows the KEGG analysis of genes differentially expressed in CRC patients also diagnosed with T2DM in comparison to those CRC patients without such co-morbidity. Please change the title of the table, as in line with the comment on the title, there is no evidence provided in the study that T2DM alters any gene expression patterns. All data presented only shows correlation, not causation. The comprehensive molecular characterisation of human colon and rectal cancer published by the Cancer Genome Atlas Network in Nature in 2012 showed that the Wnt signalling pathway was altered in more than 90% of CRC cases and that the PI3K-Akt signalling pathway was altered in more than 50% of call CRC cases. The data presented in Table 2 therefore is not surprising. Furthermore, the Hippo signalling pathway feeds into the Wnt signalling pathway and the two pathways share a large number of components (e.g. AXIN1, FZD7, WNT16 and WNT7A are even listed in the table). Likewise, CRC is often associated with an inflammatory response, which may explain the activation of TNF signalling and platelet activation. Hence, the significance and novelty of the data presented in Table 2 is not clear. All these pathways are altered in CRC with and without T2DM co-diagnosis. Could the authors please explain their rationale?
4) Discussion: The discussion lacks a critical view on the limitations of the study. In particular, the issue of cellular heterogeneity within each sample and possible inter-tumour heterogeneity between patients are not discussed. Another major issue is that the observed correlations in the data are presented as causations without any experiments to backup these claims. Since APC is inactivated in the majority of CRC cases, the relevance of altered expression of Wnt ligands such as WNT7A and WNT16 or Wnt receptors such as LRP5 and FZD7 in this context remains unclear to this reviewer, as target gene expression will occur anyway, as beta-catenin is chronically stabilised in this scenario. The authors do not take this point, which is well established in the CRC research field, into consideration. The discussion should be rewritten to incorporate the above comments.
Round 2
Reviewer 3 Report
In their revised manuscript, the authors have appropriately responded to some of my comments, namely comments 1 and 4 (partially). However, the remaining issues still remain a concern.
RE: Comment/response 2: If the goal is to specifically look at the changes in epithelial (cancer) cells, then there are many alternatives to analysing bulk tumour samples as intact pieces with non-epithelial cells 'contaminating' the data collection, such as dissociating the tissue and sorting epithelial cells out of the cell preparation. Based on the observation that the non-epithelial cell markers PTPRC, PPARG and ICAM1 (see my comments on the initial submission) are among the differential expressed genes, indicates that there are significant differences in the cell composition. This means that changes in signalling pathways such as Wnt, TNF, PI3K/Akt, etc, which are also critical regulators of haematipoietic and endothelial cell differentiation cannot be attributed exclusively to epithelial cells. The authors have not sufficiently addressed this concern.
RE: Comment/response 3: No data is provided to asses the variability/heterogeneity between groups and samples, so principal component analysis should also be performed to demonstrate lack of variability between groups.
RE: Comment/response 4: While heat maps have been provided in the revised manuscript, it seems that only 12 of the indicated 22 samples were included in the heat map in supplementary figure 1. Also, it is incorrect to state that unsupervised clustering was performed, as the genes appear to be ordered alphabetically. Could the authors please address both issues. The heat map in supplementary table 2 again shows heterogeneity in cell composition, as PTPRC (haematopoietic cell marker), COL1A1 (fibroblast marker) and other non-epithelial marker appear to show variable expression in the dataset.
RE: Comment/response 5: There is no evidence provided that T2DM is interfering with pathway modification in CRC in an epistatic/causative manner. Also, no evidence is provide that the pathway activity is actually altered, this can only been demonstrated by showing significant and robust changes in target gene expression.
RE: Comment/response 6: I agree with the authors' statement that Wnt signalling may be modulated by other routes, even in the presence of APC inactivating mutations. However, the authors do not provide any evidence for this statement. Expression of CTBP2 does not show a robust trend following the authors' hypothesis and in line with my statement above, no data are presented to show altered Wnt pathway target gene expression as a proxy of modulated pathway activity.
